# Towards Computable and Explainable Policies Using Semantic Web Standards

Henrique Santos[1],  Jamie P. McCusker[1],  John S. Erickson[1],  Alice M. Mulvehill[1],
Oshani Seneviratne[1] and  Deborah L. McGuinness[1]

[1]*Tetherless World Constellation, Rensselaer Polytechnic Institute, Troy NY, United States*

### Abstract

Policies or guidelines are defined to support decision-making in a wide range of domains. These statements describe what actions are allowed or recommended under certain conditions. The policies typically present rules that may be evaluated to generate those allowances or recommendations when encoded in machine-actionable terms. As many fields attempt to generate more computable guidelines, there is an increasing need to automatically evaluate these policies and explain results. This paper presents a novel ontology design pattern for representing policies using the OWL and PROV semantic web standards. It can be used to extend domain knowledge graphs to include representations needed to support domain-specific decision making. The encoding of policy rules using OWL restrictions over PROV entities enables the representation of common policy constructs, including subjects, actions, objects, and their attributes. This modeling can be successfully applied in a number of domains to increase inferential power and to provide better support for explaining the reasons for a given evaluation result. This is demonstrated by applying the approach with web-based tools developed for two scenarios, radio spectrum access policies, and health guidelines.

## 1. Introduction

Policies, in the context of this paper, are defined as domain-specific decision-making assets that express one or more actions that are allowed or recommended under certain conditions. Policies are commonly defined in authoritative documents in a textual (natural language) format. In contrast, there is increasing engagement by several researchers to develop computable policies [1, 2, 3, 4] to alleviate some issues associated with text-based policy authoring and evaluation.

Recent approaches [5, 6, 7] have proposed the application of semantic web standards, including the Web Ontology Language (OWL) and the Semantic Web Rule Language (SWRL), for encoding policies and guidelines. A large number of these are based on (or extensions of) the eXtensible Access Control Markup Language (XACML) [8], the "de facto" standard for representing access control policies. While well-established, it is not trivial or straightforward to leverage domain knowledge, increasingly encoded as Knowledge Graphs (KGs), within XACML constructs as a way to increase inferential power in complex domains.

This paper presents a novel ontology design pattern for representing policies and guidelines using the OWL and PROV semantic web W3C standards. Ontologies created following this "OWL+PROV" pattern extend domain knowledge graphs to include the encoding of the policy's rules in class equivalencies, expressed as OWL restrictions over domain entities. In this sense, these ontologies, in conjunction with domain knowledge from KGs, can be leveraged by OWL reasoners to classify individuals as instances of classes that represent policies if their rules are satisfied. The OWL+PROV design pattern advocates for the creation of a class hierarchy, which incrementally appends the policy's rules to support the explanation of policy evaluation results. These claims are supported by demonstrating how the approach has been applied with a web-based tool that creates ontologies in the OWL+PROV pattern for two scenarios: radio spectrum access policies (Section 3) and health guidelines (Section 4). This pattern can be applied to multiple domains while supporting the ultimate goals of computable policies: (i) to leverage domain knowledge, which is commonly not present in the policies themselves, and (ii) to explain how existing policies were used to achieve a policy-based decision.

*WOP'24: 15th Workshop on Ontology Design and Patterns, November 11–12, 2024, Baltimore, NY*

## 2. OWL+PROV: A Pattern for Representing Policies

The PROV Ontology [9] (PROV-O) provides resources for representing concepts (entities, activities, and agents) involved in producing new things and their relationships. Although the primary usage of PROV-O is to provide provenance tracking, its modeling is generic enough to be re-purposed for supporting knowledge representations that contain such concepts. The intuition behind this modeling approach is to treat "things" that are of interest (or subject to evaluation) to domain-specific policies as proposed *activities*, their associated *agents*, and their related *attributes*. Based on this premise, it is possible to classify these proposed activities as "permitted" or "denied" activities (in the case of access control policies) and as specific "recommendations" or "directives" (in the case of guidelines). When represented as individuals in RDF, proposed activities may be reasoned over and classified as instances of additional classes if they satisfy some class' equivalency requirements.

It is useful to consider the following common structural elements when analyzing policies and rules from different domains:

- *Subject:* Who/what a policy applies to (a person, a radio system, etc.)
- *Action:* What the subject is trying to accomplish, a proposed activity (a health assessment, a radio transmission, etc.)
- *Attributes:* Qualifiers for both the subject and action (a gender, a radio frequency range etc.)
- *Effect:* The outcome of a policy when its rules are satisfied (esp., a recommended action, a directive, an allowance, etc.)
- *Obligations:* Special requirements associated with an allowance (a patient having a co-morbidity, do not interfere when broadcasting, do not use more than 50% of the band, etc.)

Figure 1 shows the proposed PROV-based model for representing these policy constructs in the OWL+PROV pattern. The policy's action is represented as a prov:Activity, which is associated with the subject represented as a prov:Agent. The model seeks to reuse PROV concepts whenever possible. Location and time are attributes that may be relevant in multiple domains for expressing spatial relationships and duration, so, in this sense, the location attribute is represented as a prov:Location while the time attribute is represented using the predicates prov:startedAtTime and prov:endedAtTime. For attributes not natively supported by PROV, the model uses the Semanticscience Integrated Ontology (SIO) [10], which enables it to model objects and their attributes (measurement values, units) with the use of the sio:Attribute class and sio:hasAttribute, sio:hasValue, and sio:hasUnit predicates. The SIO ontology already contains a number of attributes, mostly for the biomedical domain, and it allows any domain to further extend it, creating additional domain attributes.

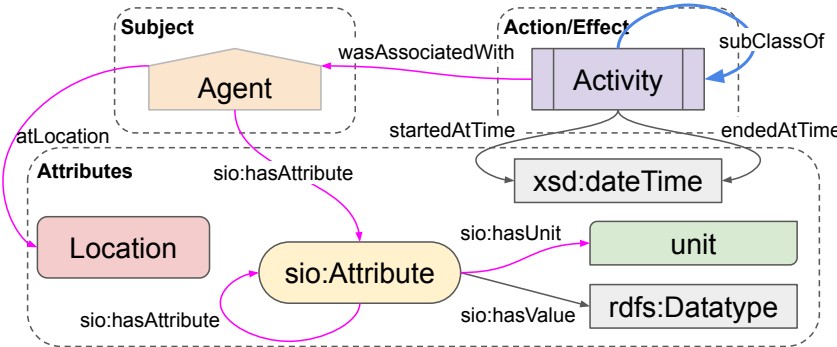

**Figure 1:** Using PROV and SIO to model policy constructs.

For domain-specific policies expressed in natural language in textual documents, it is non-trivial to effectively represent their rules in a more structured format while preserving the very same semantics contained in the original text. Depending on the domain, a policy's original text might not contain all the background knowledge required in order for it to be correctly evaluated without the aid of domain

practitioners. When interpreted by domain practitioners, textual policies can be transformed into a structured format that can then be used to create one or more logical expressions that convey a policy's rules, each leading to a unique effect, such as:

$$action \wedge subj \wedge attr0 \wedge attr1 \wedge \cdots \rightarrow effect(s)$$

These expressions combine a series of statements that qualify the action, the subject, and their attributes in a Boolean *AND* operation that, when evaluated to be *TRUE*, implies the effect(s). An OWL representation for this set of expressions can be used to classify proposed activities into classes that represent effects such as recommended, permitted, or denied. The OWL+PROV pattern advocates that a hierarchy be created where each level is represented by an OWL class that encodes one of the statements as an OWL equivalency.

### 2.0.1. Action

The policy's *action rule* is represented as a subclass of prov:Activity, and it is the top-level class in the hierarchy. This class does not specify an OWL equivalency as it expects "things" that are subject to evaluation to directly refer to the action they want to accomplish.

### 2.0.2. Subject

The policy's *subject rule* is represented as a subclass of the Action class. This class specifies an OWL equivalency to append the subject rule to the action rule, as seen below. The equivalency is defined as an intersection of the Action class and the OWL constraint on the prov:wasAssociatedWith property.

```
1  ...
2  EquivalentTo:
3      Action and (wasAssociatedWith some Some_Subject)
4  SubClassOf:
5      Action
```

Listing 1: OWL equivalency for the *subject rule* in Manchester syntax.

### 2.0.3. Attributes

The policy's attribute rules are represented as classes that further extend the hierarchy as sub-classes. As seen below, these classes specify equivalencies defined as an intersection of the immediate super-class and the constraint for the attribute rule being encoded.

```
1  ...
2  EquivalentTo:
3      Subject and
4      wasAssociatedWith some (hasAttribute some Some_Attribute)
5  SubClassOf:
6      Subject
7  ...
8  EquivalentTo:
9      Attribute and
10     wasAssociatedWith some (atLocation some Some_Location)
11 SubClassOf:
12     Attribute
13 ...
14 EquivalentTo:
15     Location and
16     startedAtTime value "some_Value"
17 SubClassOf:
18     Location
```

Listing 2: OWL equivalency for the *attribute rules* in Manchester syntax.

### 2.0.4. Effect

The pattern represents policies' effects as activity classes that are satisfied when proposed activities are reasoned to belong to specific policies. When the OWL class hierarchy completes the representation of a logical expression, it must define the policy's effect that is associated with the expression. This definition is performed by stating that the last class in the hierarchy is a subclass of the class that represents the effect, as seen below.

```
1 ...
2 SubClassOf:
3   Rule, Effect
```

Listing 3: *Effects* represented as classes in Manchester syntax.

### 2.0.5. Obligations

Obligations are requirements that can be specified alongside effects. They should be accomplished for the effect to be valid. Much like an effect, an obligation can be seen as an activity class that needs to be satisfied for the effect to be valid as well, and the pattern represents obligations in a similar way, as seen below.

```
1 ...
2 SubClassOf:
3   Rule, Effect, Obligation_1
```

Listing 4: *Obligations* represented as classes in Manchester syntax.

This hierarchy of OWL classes will maximize the reuse of encoded rules. It is common for a policy to contain a set of logical expressions that share some of the rules. Even among different policies, the logical expressions might contain similar statements that could be reused when encoded in OWL. Another advantage of this hierarchical approach is that it supports the explanation of evaluation results by traversing the OWL graph to find rules that were not satisfied.

## 2.1. Variations of the Pattern

In some domains, advantages might exist for changing the order in the class hierarchy, or even interlacing policy constructs to better support the implementation of the pattern. As an example, attribute-based access control (ABAC) policies [11] might benefit by having the access-controlled resource attribute evaluated earlier and, therefore, "higher" in the hierarchy before the subject. In access control applications, a common set of resources may be associated with different subjects and additional attributes. Having the resource evaluated earlier allows this class to be reused by any subject or additional attribute class, thereby minimizing the number of duplicate rules.

In other cases, the construction of this OWL class hierarchy might not directly benefit the implementation of the pattern, because either the explanation of results should be done in a different way than traversing the OWL graph or there are no requirements for reusing previously-encoded rules. In this sense, the OWL class equivalencies can be created in lesser numbers, by combining multiple attributes, subjects, and even actions, in a single class. We call this approach the "flat" representation of policies.

## 3. Use Case 1: Dynamic Spectrum Access

The OWL+PROV pattern was applied in the construction of an ontology for representing and evaluating spectrum access control policies. This is the core feature of the Dynamic Spectrum Access (DSA) Policy Framework [12], a semantically-enabled system for managing machine-readable policies. In this domain, policies regulate the use of specific partitions of the radio spectrum by permitting or denying the usage of a specified frequency (or frequency range) based on the who/what is requesting usage (a

| RuleID | Parsed logical rule | Requester | Affiliation | Frequency | Location | Effect | Obligation |
|---|---|---|---|---|---|---|---|
| US91 | IF RefFreq is ε ( ≥ 1755 MHz AND ≤ 1780 MHz) then the following provisions apply | | | | | | |
| US91-1 | IF TR is AWS AND RefFreq is ε ( ≥ 1755 MHz AND ≤ 1780 MHz) AND (TR has successfully coordinated on a nationwide basis prior to operation, unless otherwise specified by Commision rule, order , or notice) THEN TR is Primary | AWS | non Federal | 1755 MHz - 1780 MHz | | Permit with Obligation | TR has successfully coordinated on a nationwide basis prior to operation, unless otherwise specified by Commision rule, order , or notice and TR is Primary |

**Figure 2:** Spreadsheet excerpt showing the NTIA Redbook US91 policy capture.

device, an organization, a system, etc.), their affiliation (federal, non-federal), and where an instance of the requester type is spatially located.

During the *policy capture* process, Rensselaer Polytechnic Institute (RPI) collaborated with DSA domain experts from Capraro Technologies Inc. and LGS Labs of CACI International Inc. to select and analyze English language, text-based policies from the National Telecommunications and Information Administration (NTIA) Redbook [13]. It was observed that the text for many policies is equivalent to a logical conditional expression, e.g., *IF* some device wants to use a frequency in a particular frequency range *AND* at a particular location, *THEN* it is either *PERMITTED* or *DENIED*. More complex policies contain a set of expressions, with each conditional expression focused on a particular attribute, e.g., a device type, frequency, frequency range. Each conditional expression can be understood as a sub-policy (e.g. a policy that further constrains an existing policy). The spreadsheet displayed in Figure 2 contains an example of a complex policy from the NTIA Redbook, called US91. Due to space constraints, we have omitted some of the sub-policies for US91 and the columns that document policy metadata and provenance, including the original text, source document, URL, and page number.

The elements of the logical expression are further expressed in the columns as attribute-value pairs, e.g., Requester = AWS, and mapped to the PROV model:

- *Requester* (prov:Agent): the device requesting usage of the spectrum
- *Action* (prov:Activity): the purpose for the usage of the spectrum
- *Affiliation* (sio:Attribute): the affiliation of the requester: Federal or Non-Federal
- *Frequency* (sio:Attribute): the frequency range or single frequency being requested for use
- *Location* (prov:Location): location(s) where the policy is applicable
- *Effect* (prov:Activity): the allowance or denial a policy expresses, if the rule is satisfied (Permit, Deny, Permit with Obligations)
- *Obligations* the list of obligations the requester needs to comply with in order to be permitted

Based on the expressed attribute-value pairs, an OWL class hierarchy was created, as shown in Listing 5, with each class directly representing a single sub-policy as expressed in the policy capture spreadsheet (Figure 2). As the first class in the hierarchy, US91 extends the action class Transmission (line 14) and appends the frequency range rule for the US91 policy (lines 5-12). In its turn, the US91-1 sub-policy is represented by a class that extends US91 and appends the Requester and Affiliation rules (lines 19-20). This sub-policy expresses the Permit effect; therefore, US91-1 is a subclass of PermittedActivity.

```
1  ...
2  Class: US91
3    EquivalentTo:
4      Transmission and
5      (wasAssociatedWith some (hasAttribute some
6        (FrequencyRange
7         and (hasAttribute some
8             (FrequencyMaximum and
9               (hasValue some xsd:float[<= 1780.0f]))))
```

```
10          and (hasAttribute some
11              (FrequencyMinimum and
12                  (hasValue some xsd:float[>= 1755.0f]))))))
13      SubClassOf:
14          Transmission
15
16  Class: US91-1
17      EquivalentTo:
18          US91 and
19          (wasAssociatedWith some AdvancedWirelessService) and
20          (wasAssociatedWith some (hasAttribute some NonFederal))
21      SubClassOf:
22          US91, US91-1-Obligation,
23              PermittedActivity
```

Listing 5: OWL expression of part of the US91 policy in Manchester syntax.

To demonstrate how this policy is evaluated we use the following example: an instantiation of the model in Figure 1 as a request expresses that a requester of type AdvancedWirelessService (a prov:Agent) wants to make a Transmission (a prov:Activity) using the 1755-1756.25MHz frequency range. The requester is at the location specified by the coordinates POINT(-114.23 33.20) and has the Non-Federal attribute. Because the request specifies a Transmission, the requirement for this class is satisfied. Next, the requested frequency range falls within the range constrained by the US91 class, satisfying this class as well. The agent type in the request is an AdvancedWirelessService with the Non-Federal attribute, satisfying the US91-1 class. Therefore, the request's activity is reasoned to be an instance of US91 and US91-1. As US91-1 is a subclass of PermittedActivity, the attempted transmission of the request is also an instance of PermittedActivity.

The DSA framework [12] contains an evaluation engine that takes a set of transmission requests as input and outputs the assigned effect, a list of obligations, and a list of reasons to explain the effect for each request. For this explanation, the implementation employs a mix of OWL reasoning and graph traversal to identify classes that were not satisfied during reasoning. The unsatisfied rules are then presented as reasons.

## 4. Use Case 2: Health Guidelines

In healthcare, clinical practice guidelines are systematically developed statements to assist practitioners and patients. There are guidelines for various medical disciplines such as allergy, cardiology, family medicine, gastroenterology, men's health, women's health, neurology, oncology, pediatrics, psychiatry, etc. There is immense value in encoding these guidlines in computable formats to supplement clinical decision support systems [14, 15, 16], integrate connected health applications [17], and provide semantics-driven dietary recommendations [18, 19, 20]. Symbolic reasoning processes that are enabled by such semantically-rich explainable policies are shown to effect patient persuasion and education towards better health outcomes [21, 22].

In this section, we will demonstrate the application of the OWL+PROV to diabetes guidelines [23] by leveraging the Guideline Provenance (G-Prov) ontology [24]. These guidelines were developed as part of the Health Empowerment by Analytics Learning and Semantics (HEALS) project[1] at RPI. In the HEALS project, we primarily focused on the semantic modeling of recommendations from lifestyle interventions and pharmacological treatment guidelines. The "Lifestyle Management" position statement [25] defines a set of policies for supporting medical nutrition treatment for such patients. These policies are categorized by topics (such as "Dietary Fat" and "Alcohol"), and each policy provides an "Evidence Rating" stating how strongly various existing studies support the policy. Here is an example from the guideline:

*"Adults with diabetes who drink alcohol should do so in moderation (no more than one drink*

---

[1]https://idea.rpi.edu/research/projects/heals

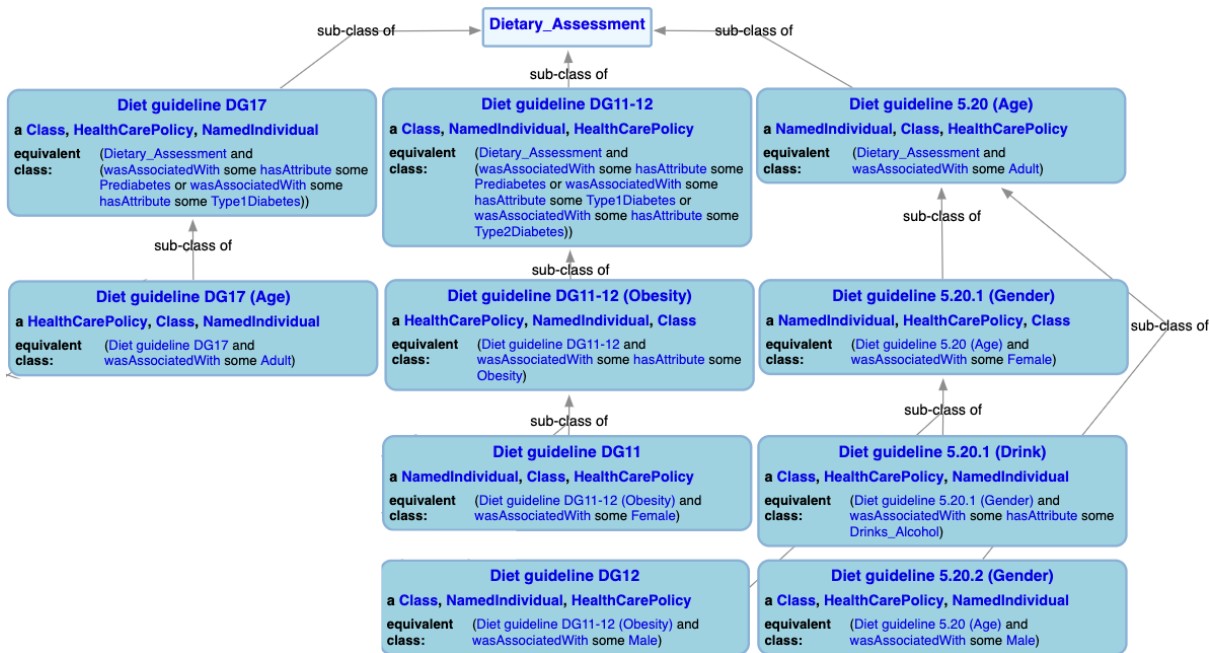

**Figure 3:** Selected dietary recommendations from the guideline modeled using OWL+PROV.

*per day for adult women and no more than two drinks per day for adult men). Evidence Rating = C."*

In the above policy example, we can identify the recommended action, which is to drink alcohol in moderation. Additional information is provided regarding what moderation means in terms of the quantity. The rules (conditions) for when this recommendation should be valid can be identified. These include (i) the person is an adult, (ii) the person has diabetes, and (iii) the person drinks alcohol. In addition, the recommendation specifies different actions for males and females, so each gender is also part (a condition) of the policy's rules. The supporting evidence rating has a "grade" C, which is defined as evidence from poorly controlled or uncontrolled studies. Such a guideline would be weighed lower by a health practitioner compared to a health guideline that has an evidence rating of either A or B.

When modeling health guidelines, we observe a similar pattern as radio spectrum (Section 3), even though they are vastly different domains. Health guidelines (referred to as policies within this paper) encapsulate observations (a set of conditions) and one or more recommended actions derived from practitioner expertise and/or the published results of studies and research. The policies we modeled as part of the HEALS project recommend evidence-based recommendations for a diabetic patient, based on their lifestyle, diabetes condition, ongoing treatment, and diet.

Figure 3 shows a section of some of the diabetes guidelines that we have modeled, focusing on lifestyle interventions. These terms were modeled in a domain ontology and then referred by the policies, including temporal pattern descriptors (i.e., $\mathrm{Drinks\_Alcohol}$).

In addition, we represent the evidence rating as a class (i.e. $\mathrm{GradeC}$) and assert the last class in this policy's hierarchy to be a subclass of it. By doing so, we enable a similar precedence evaluation approach as we did in the radio spectrum domain.

A demo video outlining the usage of these guidelines is available[2] if one prefers to get a better understanding on the background and the motivation behind the health guidelines we have modeled.

---

[2]https://foodkg.github.io/demo.html

# 5. Evaluation

The OWL+PROV ontology design pattern was evaluated using Thorn's criteria [26] for model quality based on its changeability, reusability, formalness, mobility, correctness, and usability. For this evaluation, we have used ontologies created for the radio spectrum domain, as we had access to a comprehensive domain KG to support them and stakeholders to validate policies' contents. In general, the OWL+PROV approach does very well, although, without support tools, usability issues can arise due to the initial counterintuitiveness in the modeling process.

*Changeability* is the ability to evolve the model while maintaining the uses of previous versions. During the use of this approach, we have identified the following ways in which the model changes:

- New policy attributes: The Affiliation attribute surfaced during the course of the project. To accommodate the new attribute, we first represented Affiliation as a new attribute in the domain ontology, and then, we created a new class in the class hierarchy of affected policies with the appropriate affiliation rule. The existing rules remained unaffected.
- Novel evaluation rules of existing attributes: At first, we had the Location attribute being evaluated as a calculation of some coordinate being within some known polygon. Later, some new policy rules required the location attribute to be evaluated in terms of distances. To allow this, we have created a new class in the class hierarchy of policies containing such rules that would now contain *distance* rules instead of *within* rules.

*Reusability* is the ability to reuse (parts of) the model when evolving or developing other models. The reusability of this ontology design pattern is very high. We originally developed OWL+PROV to express policies around radio spectrum allocation and were able to re-apply the approach to a relatively diverse set of other areas, including medical treatment guidelines. Further, these reuses have helped to inform each other, as we were able to see a path to modeling obligations consistently by analyzing the correct approach in deontic logic.

The *formalness* of a model is the ability to manage the model in a formalized manner. OWL+PROV maps neatly onto the exact truthmaker semantics expressed in [27] and provides an integration point among multiple modal logics by denoting more than one set of top-level classes of activities. These semantics are more precise than classical formulations of modal logics by explicitly mapping them to activities (thereby restricting the universe of discourse) and providing an OWL-compatible set-theoretic basis for distinguishing between kinds of activities and individual acts. Additionally, models developed using OWL+PROV use formal representations rigorously enough to actually evaluate actions based on classifying them into specific policies. As shown in Section 3, moving from a data model representation based on XACML to OWL+PROV enabled us to directly evaluate policies using an existing OWL reasoner.

The *mobility* of a model is the ability to be moved, transferred, and integrated with other systems. We have used two different approaches to evaluate the mobility of ontologies created in the OWL+PROV pattern. Common Logic [28] is an ISO standard that defines a family of logic languages that enable the interchange of knowledge among computer systems. We have used the Common Logic Interchange Format (CLIF) as a way of serializing the OWL representation of policies. This approach was implemented in [12] as a way of exporting existing policies.

*Correctness* is the correspondence (or mapping) between the model and the modeled artifacts. For radio spectrum policy, prior work in [12] describes the representation coverage for radio spectrum policy as published. We have tested our policy framework on a subset of the United States NTIA radio spectrum policies [13], using example activity requests that produce the expected classifications of those requests as permitted or prohibited activities.

*Usability* is the ease of effectively communicating a model to new users and the ability to align the model with the users' "mental models," or the users' conception of how the representation works, thereby minimizing learning curves. We found that users do not always share the same initial assumptions about what a "policy" is. However, once they understood that a policy is a way for expressing kinds

of actions that might take place (and thereby be allowed or not), the users with an understanding of OWL were able to quickly build policy rules directly in OWL using existing semantic web tools like Protege. The users that did not have ontology development training but did understand the domain policies were able (with some training) to adopt some new terminology, e.g., permit and deny and use the tools and domain-specific languages that we built to accurately express policies, while the tools automatically translated the policies into the appropriate OWL class definitions.

## 6. Related Work

Existing literature about semantic approaches to policy representation includes Kirrane [7], which offers a comprehensive survey of access control models, well-known policy languages, proposed frameworks that utilize ontologies and/or rules to express policies, and categorization of policy languages and frameworks against access control requirements. Thi [29] proposes an OWL-based extension to the eXtensible Access Control Markup Language (XACML 3.0) [8] to support a generalized context-aware role-based access control (RBAC) model providing spatio-temporal restrictions and conforming with the NIST RBAC standard [30]. Their work augments the XACML architecture with new functions and data types.

Kolovski [31] maps the web service policy language, WS-Policy [32], to the description logic fragment species of OWL and demonstrates how standard OWL reasoners can check policy conformity and perform policy analysis tasks.

Garcìa [33] and Finin [34] offer important contributions to how end-to-end usage rights and access control systems may be implemented in OWL and RDF. Garcìa proposes a "Copyright Ontology" based on OWL and RDF for expressing rights, representations that can be associated with media fragments in a web-scale "rights value change." Finin describes two ways to support standard RBAC models in OWL and discusses how their OWL implementations can be extended to model attribute-based RBAC or, more generally, ABAC. OWL-POLAR [35] is a model based on OWL DL designed to manage and analyze policies as soft constraints for agents. It enables the representation of norms and their activation conditions using conjunctive semantic formulas and employs SPARQL queries for reasoning about policy conflicts and resolution. By leveraging ontology consistency checkers, it can detect conflicts among norms and determine resolutions, though it does not address temporal aspects.

Fenz [36] supports complex decision-making with an approach that translates technical decision options into a language that is understood by relevant stakeholders. This is accomplished by following a proposed 5-step ontology engineering method. Ontologies created with this method model relevant problem parameters as RDF resources (including potential solutions for the problem and physical entities of the domain), and contain description logics statements as class equivalencies capable of classifying the model entities into potential solutions. While this work has similarities with our approach, as we encode policy rules as class equivalencies for classifying proposed activities into recommendations, it doesn't offer support for explaining evaluation results and for reusing existing rules.

The Open Digital Rights Language (ODRL) [37] provides an ontology that is especially applicable for modeling policies that embody agreements concerning intellectual property, especially copyrighted works. Instead, we chose to base our foundational modeling on XACML, which, as noted, has a long implementation history in cross-domain access control. Both models consider permissions and obligations, and ODRL arguably provides more flexibility and extensibility for modeling agreements such as licenses, which we will consider for future policy-based spectrum management work. The scope of our current work did not permit the expression and evaluation of spectrum licenses or agreements; future work will include this, and we will look to work based on ODRL for inspiration, especially for guidance in logically modeling agreements.

With respect to medical guidelines, some of our previous work has demonstrated how a semantic technology approach can be used in characterizing disease (breast cancer) based on newly emerging criteria using OWL [38] and representing the provenance of specific disease guidelines where we demonstrate the application of the W3C provenance ontology in ADA guideline annotation [24].

Our approach combines OWL, PROV-O, and the HermiT OWL reasoner with an ontology, represented as a knowledge graph, to support the representation of decision-making policies in multiple domains. Relevant related research is described in Dundua [39], in which OWL is used for modeling and analyzing access control policies, especially ABAC, and the integration of the ABAC model into ontology languages is considered. In addition, Sharma [40] describes how OWL can be used to formally define and process security policies that can be captured using ABAC models. This work demonstrates how models, domains, data, and security policies can be expressed in OWL and how a reasoner can be used to decide what is permitted.

## 7. Conclusion

The growing creation and adoption of domain-specific knowledge graphs over the past decade, both in industry and academia, has resulted in an increased number of domain entities being represented in such structures. In many knowledge graph (KG) applications, the entities include domain terms that are referred to in textual policies (and their attributes). Being able to leverage such knowledge in policy evaluations can allow for the creation of more intelligent evaluation processes, including increased evaluation power due to the expressiveness of KGs and better techniques for explaining the reasons for a given evaluation result. In addition, since policies usually do not contain all of the required background knowledge to allow for their correct evaluation, as demonstrated in the paper, the OWL+PROV pattern enables this domain knowledge in KGs to be leveraged by allowing the OWL constraints that represent a policy's rules to refer to domain entities in domain KGs.

This policy representation approach builds on previous work by matching the cross-domain policy expression semantics of XACML. It extends these semantics with the capacity to express rich spatio-temporal restrictions, enabling the implementation of a wide variety of attribute-based policies across domains. It leverages background knowledge from domain-specific knowledge graphs that are structured with a domain-derived ontology, enabling the inference of policy applicability based on attributes and constraints. Our approach uniquely conceptualizes policy requests as PROV activities and request evaluations as realizations. Finally, the OWL+PROV pattern allows a novel reasoner-based explanation in request evaluation results, enabling domain policy developers to understand the precise reasons for policy decisions.

In terms of limitations, the domain knowledge is currently used "as is," and policy evaluations will be constrained by the quality of the domain graph. In the DSA and health domains, we have encountered performance issues with OWL reasoning due to the number of axioms involved in the process. To alleviate this, we first removed a number of axioms from the supporting ontologies (PROV and SIO) that we judged were not relevant to the policy evaluation process, which drastically reduced the reasoning time. In an additional effort to further reduce the required reasoning time, we modified our implementation to segment the graph containing individuals subject to the evaluation into smaller graphs and then used parallel processing to reason over each graph. The results were combined after the conclusion of each individual processing thread.

A web-based tool [41] that leverages domain knowledge graphs to allow the creation of policies using the OWL+PROV pattern is available.

## Acknowledgments

This work is partially funded in support of National Spectrum Consortium (NSC) project number NSC-17-7030 and partially supported by IBM Research AI through the AI Horizons Network.

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
