# OpenReview forum: "Towards Computable and Explainable Policies Using Semantic Web Standards"
_swsa.semanticweb.org/ISWC/2024/Workshop/WOP — WOP 2024 Oral_

### Official Review · Reviewer_PVhV · 2024-08-26
**Review of Paper 5: Towards Computable and Explainable Policies Using Semantic Web Standards**

**Rating:** 7
**Confidence:** 3

**Review:**

### Overview:
In this paper, the authors propose a domain-independent ontology design pattern for policy modeling based on OWL and PROV. They emphasize the need to harmonize efforts to automated evaluation in the face of the increasing generation of computable policies and guidelines. As a module that can be used to extend existing domain knowledge graphs (KG) with decision information and add inference power for automatic evaluation and explanation of policy application, the paper demonstrates its applicability in two use cases: radio spectrum access policies and health care guidelines.

### Practical Utility and Reusability of the Pattern (within a Community):
The pattern makes it possible to go from natural language-based, textual policies and guidelines to a structured representation. For the use cases in section 3 and 4, there is already an example for capturing policies. For further emphasis of the pattern's value for the field, it could be helpful to further enrich the description of this process and combine this with examples from a KG, showing how the pattern affects this transformation, how this is combined with a KG, and how the pattern aids in enhancing decision-making across various domains.

### Relevance of the Problem Addressed by the Pattern:
OWL+PROV addresses a relevant issue in this area, particularly with regard to the increasing popularity of automated approaches to policy evaluation, decision making, and the explainability of those decisions. The combination of OWL and PROV addresses the need for a modular framework that can capture the complexity of policy management. Recognizing the need for a domain-independent, modular framework for representing textual policies in a structured, KG-interoperable manner, and for making policy explanations both human- and machine-understandable, the authors provide the OWL+PROV pattern as an extension to existing modeling approaches such as XACML. With respect to existing work on this topic, the authors emphasize the need to go beyond XACML definitions of context-aware policies – which can be challenging and not straightforward – and to allow the reuse of existing rules, taking into account structured domain knowledge for evaluation and explanation of results.
This addresses the need to efficiently encode policies and reduce redundancies in their representation through the hierarchical organization of classes and the sharing of logical expressions among them. In particular, as the use cases show, this can be beneficial when working on similar policy challenges. With the addition of PROV-O to the mix, the pattern helps make decisions more transparent by representing activities (e.g., policy requests) and agents (e.g., users or systems), and inferring relationships based on the reasoning of encoded policy rules, also based on the domain data in the KG. Thus, the OWL+PROV pattern addresses a relevant problem in the domain, proposing an efficient way to encode diverse policies.


### Best Practice within a Community:
As shown in the Related Works section, the authors consider the existing work in the field and the approaches available for expressing policies in a semantic web environment. The proposed pattern is an extension of the policy representation of XACML, a de facto best practice in the field, which provides cross-domain semantics for policy expression. OWL+PROV is a reconciliation of existing approaches with more expressive semantics, adding the ability to provide reasoner-based explanations for policy evaluation results. By enabling context-aware policy evaluation and more transparent insights into the specific reasons behind policy decisions, the OWL+PROV pattern is well positioned to become a best practice in various communities (as shown in the use cases). It would further emphasize this fact with a more detailed revision of what KGs with this type of pattern as a backbone would look like in comparison to more ‘traditional’ modeling approaches.


### Real-World Use:
The authors demonstrate the real-world use of the OWL+PROV pattern in two use cases: 1) radio spectrum access policies and 2) health care guidelines. For 1), the pattern is applied to radio spectrum access management, and a detailed description of the workflow from textual policy to ontology representation is given. Use case 2) shows the applicability of OWL+PROV in the health sector, where it helps to ensure compliance with health regulations and recommendations.
For both use cases, the application of the pattern shows its applicability in real-world scenarios in different domains.


### Overall evaluation, questions, and recommendations:
* What were the specific challenges you encountered in ‘recycling’ PROV for your purpose? Did you find the reuse of its classes and properties challenging in any way, and if so, would you see this as a problem that could make the pattern difficult for potential reusers to understand? Perhaps you could address this further in the use case scenarios or conclusion?
* In the Related Work and Conclusion sections, you mention that you encountered scale and performance issues with OWL reasoning when implementing the OWL+PROV pattern. How does this affect your modeling approach (except for removing axioms or reasoning over subgraphs)? Have you considered changing the hierarchical structure of your ontology (which you mention also in the use cases, e.g. section 3)? Also: Perhaps a visual representation of exemplary KGs from the use cases using your pattern (similar to Figure 1 but with A- and T-BOX) could underscore the complexity (but also the strength of being able to represent such complex policies) of OWL+PROV?
* This might be more of a stylistic choice: Personally, not being from this exact field, reading the Related Works section at the end was a challenge because I was lacking some information on the existing approaches for policy and guideline ontology modeling. In the context of a more ‘ontology-oriented’ workshop, where many people may not be experts in the field, putting it after the introduction might make it easier for them to understand the strength of your OWL+PROV pattern.

The pattern presented in this paper contributes to the development of ontologies and related conceptualizations in the context of policy representation, automation, and evaluation. It is an innovative approach that goes beyond XACML, adding the perspective of background knowledge for correct policy evaluation and decision making with enhanced transparency. The paper also provides an evaluation part, for which the authors used Thorn’s criteria. This already gives a first hint of strengths and weaknesses of the pattern. Answering the questions from above would add even further to this.

---

### Official Review · Reviewer_mTMU · 2024-09-06
**Reusing PROV as a new pattern for policies**

**Rating:** 7
**Confidence:** 4

**Review:**

The paper presents a pattern for representing policies based on the PROV-O, and exemplifies its usage in two distinct domains. Additionally it is assessed in relation to a set of quality criteria.

The paper reads very well and is quite interesting. The work is not extremely novel, since it is mostly an adaptation of the PROV-O, but in my opinion novelty is anyway not a good measure for ontology design patterns, since they are supposed to capture the essence of some “common pattern” rather than invent something new necessarily. In fact, I really appreciate this paper because of the attempt to reuse and repurpose existing work, rather than applying the very common argument “the ontology does not do exactly what we want so we created a new one”. I believe that this kind of work is what is really needed to make the current ecosystems of ontologies work - find the commonalities, rather than look for the differences.

Still, the paper could benefit from some minor improvements. For instance, the related work section mainly lists a number of alternative approaches, without contrasting them against the currently proposed pattern. A few sentences are there to explain why existing models and approaches are not sufficient, but I would expect each mentioned work to be compared (theoretically) to the current proposed pattern, to explain differences, and show the “white spots” that this work fills.

Starting from the beginning of the paper, it is also not so clear to me why the approach is called OWL + PROV? Isn’t PROV-O already that, i.e. the OWL version of PROV? Since SIO seems to be the other ontology reused, then it might even have been more intuitive to call it PROV + SIO?

While I appreciate the details of section 3, it feels slightly unbalanced that section 4 is then very short and has no concrete examples of the formalisation. It would have been very nice to have one example there as well, corresponding to one of the same types of expressions as in Listing 5, but for the medical domain. Potentially also additional examples (for both cases) could be provided in additional online supplementary material.

Minor details:
I am not sure I understand why a patient having a co-morbidity is an obligation (on page 2).

I would appreciate a bit better motivation of why SIO is reused. It is merely mentioned, but not motivated. Are there alternatives? I guess units for instance are also in QUDT?